# Features of *Mycobacterium bovis* Complete Genomes Belonging to 5 Different Lineages

**DOI:** 10.3390/microorganisms11010177

**Published:** 2023-01-11

**Authors:** Ciriac Charles, Cyril Conde, Fabien Vorimore, Thierry Cochard, Lorraine Michelet, Maria Laura Boschiroli, Franck Biet

**Affiliations:** 1Animal Health Laboratory, National Reference Laboratory for Tuberculosis, Paris-Est University, French Agency for Food, Environmental and Occupational Health and Safety (ANSES), CEDEX, 94701 Maisons-Alfort, France; 2Infectiologie et Santé Publique (ISP), Institut National de Recherche pour L’agriculture, L’alimentation et L’environnement (INRAE), Université de Tours, UMR 1282, 37380 Nouzilly, France; 3Laboratory for Food Safety, Unit of ‘Pathogenic *E. coli*’ (COLiPATH) & Genomics Platform ‘IdentyPath’ (IDPA), ANSES, 94701 Maisons-Alfort, France

**Keywords:** *Mycobacterium bovis*, mammalian tuberculosis (TB), complete de novo assembly, transmission, pangenome

## Abstract

Mammalian tuberculosis (TB) is a zoonotic disease mainly due to *Mycobacterium bovis* (*M. bovis*). A current challenge for its eradication is understanding its transmission within multi-host systems. Improvements in long-read sequencing technologies have made it possible to obtain complete bacterial genomes that provide a comprehensive view of species-specific genomic features. In the context of TB, new genomic references based on complete genomes genetically close to field strains are also essential to perform precise field molecular epidemiological studies. A total of 10 *M. bovis* strains representing each genetic lineage identified in France and in other countries were selected for performing complete assembly of their genomes. Pangenome analysis revealed a “closed” pangenome composed of 3900 core genes and only 96 accessory genes. Whole genomes-based alignment using progressive Mauve showed remarkable conservation of the genomic synteny except that the genomes have a variable number of copies of IS*6110*. Characteristic genomic traits of each lineage were identified through the discovery of specific indels. Altogether, these results provide new genetic features that improve the description of *M. bovis* lineages. The availability of new complete representative genomes of *M. bovis* will be useful to epidemiological studies and better understand the transmission of this clonal-evolving pathogen.

## 1. Introduction

Mammalian tuberculosis (TB) is a zoonotic disease mainly due to *Mycobacterium bovis* (*M. bovis*). Within *M. bovis*, four major clonal complexes were defined by the lack of certain specific regions, single nucleotide polymorphism (SNP), and genetics signatures in the DR region [1,2,3,4]. These four groups are the European 1 clonal complex (Eu1) mainly present in the British Islands, and the former British empire colonies [4], the European 2 clonal complex (Eu2), dominant in the Iberian Peninsula [3], the African 1 clonal complex (Af1) present in Mali, Cameroon, Nigeria and Chad [2], and the African 2 clonal complex (Af2) mostly found in East Africa [1].

The generalization of genome sequencing in the last years has made it possible to obtain several thousands of short-reads whole genome sequences of *M. bovis* [5,6,7,8,9,10]. These data are useful to propose *M. bovis* classification [7,10,11], such as that of Zwyer et al. proposing to classify this species into eight sublineages named La1.1 to La1.8 [11]. The *M. bovis* French diversity has been divided into nine clusters (Cluster A to I, cluster D represents Eu1 and cluster F represents Eu2), which have been defined by specific SNPs, particular signatures in the DR region and in certain VNTR loci [7]. Strains belonging to cluster A, cluster I, and cluster C provoked the majority of outbreaks detected in France in the last 20 years [7,12,13,14]. New complete genomes belonging to these different groups could help improve the description of *M. bovis* clusters defined previously. Indeed, most sequenced genomes are drafts. These genomes are incomplete and contain indels which can bias genetic structure studies or pangenomic studies [15]. Genome sequencing using long-read technologies now makes it possible to correct these errors and complete the genome at a lower cost [16].

Until recently, only AF2122/97 (NC_002945.4), the complete genome of a Eu1 field strain isolated in the UK, was used as a reference in whole genome SNP (wgSNP) studies [17,18]. Even though this reference genome is well adapted for epidemiological studies where Eu1 strains are common [11,19,20,21,22,23], it could be less fitted for studies in France and other mainland European countries where strains belonging to this clonal complex are not frequent [3]. Recently, a new complete genome, Mb3601, has been published [24]. Mb3601 was obtained by combining short-reads (Illumina) and long-read (PacBio) sequencing technologies. This new genome is specific to one of the most widespread genotypes in France in the last years, SB0120-CO [25]. The study of this complete genome has highlighted the presence of a significant number of IS*6110* copies and the presence of several indels in its genome. Its description led to the proposal of a new clonal complex, European 3, to replace Cluster I. 

The aim of this work was to obtain new complete genomes that represent *M. bovis* lineages identified in France selected among the main genotypes responsible for TB in the last years to refine our genomic knowledge of *M. bovis* via pangenome analysis and to provide better resolution of the phylogeny needed to study the epidemiology, transmission, and evolution of this clonal pathogen.

## 2. Materials and Methods

### 2.1. Mycobacterium bovis Strains

A total of 10 strains that cover all *M. bovis* lineages identified in France and representing the main genotypes responsible for TB outbreaks were selected from the National reference laboratory strain collection (Table 1). These strains were grown in Middlebrook 7H9 + ADC enrichment supplement as described previously [7].

### 2.2. Additional Genomes 

To improve pangenomic and SNP studies, 86 genomes representative of the *M. bovis* French diversity obtained by Illumina technology in a previous study [7] and 2 complete genomes (Mb3601 and AF2122/97) were included [18,24].

### 2.3. DNA Extraction

DNA extraction was performed on 40 mL of stationary phase culture using CTAB and phenol chloroform [7,26,27]. DNA concentration was measured with Qubit 2.0 Fluorometer (Thermo Fisher Scientific, Rodano, MI, Italy) using the “dsDNA BR Assay” kit (Thermo Fisher Scientific). For MinION sequencing, DNA qualities were checked with Nanodrop and DNA integrity was checked with an Agilent 4200 Tapestation. For Illumina sequencing, control quality was performed by Genoscreen (Lille, France) by SybrGreen assay (Thermofisher scientific) and qualitatively controlled by agarose gel electrophoresis.

### 2.4. MinION Sequencing

Each DNA sample was purified with AMPure XP beads (Beckman coulter, Villepinte, France). Samples were adjusted to 2 µg in 50 µL with Qubit (dsDNA BR Assay) quantification and diluted with DNAse, RNAse-free water. The MinION library was prepared according to Nanopore’s protocol “Native barcoding genomic DNA (with EXP-NBD104, EXP-NBD114, and SQK-LSK109)” (Version: NBE_9065_v109_revV_14Aug2019). DNA pool of 324 ng was loaded on an (R9.4.1) flow cell and was sequenced with Oxford Nanopore MinION sequencer for 48h (Appendix A).

### 2.5. Illumina Sequencing

Nextera XT sequencing libraries were generated with the “Nextera XT DNA Library Prep” kit according to the supplier’s recommendations, except for the equimolar pool preparation (GenoScreen optimization). Whole genome paired end 2 × 150 bases pairs (bp) sequencing was performed using Illumina MiSeq technology by Genoscreen (Lille, France) (Appendix A). To avoid PCR overrepresented fragments during the library preparation, the paired-end FASTQ files were filtered, leaving only one pair of replicated short reads.

### 2.6. Genome Assembly Method

The quality of sequencing reads was evaluated using FastQC (Version 0.11.9) (https://www.bioinformatics.babraham.ac.uk/projects/fastqc/ (accessed on 10 August 2021)). Reads were trimmed with Sickle (Version 1.33) (https://github.com/najoshi/sickle (accessed on 10 August 2021)) using a quality Phred-score of Q20 [28,29] and Nanoflit (version 1.1.0) [30] with -q 8 and -l 500 options for long-read sequencing. For each genome, Trycycler [31] subsample was used to perform 12 different read sets from initial long reads. These data were assembled with Flye (version 2.8.3-b 1695) [32], Raven (version 1.50) [33], and Unicycler (version v0.4.9b) tools [34]. These tools give different assemblies, and a consensus assembly was obtained for each strain using Trycycler (version v0.5.0) [31]. The consensus assemblies were polished with Medaka (version 1.4.3) (https://github.com/nanoporetech/medaka (accessed on 10 August 2021)) to create consensus sequences and variant calls from long-read sequences against the previously assembled genome. Assemblies were corrected using short-reads and Pilon (version 1.24) [35]. Pilon was executed until 2 runs returned no corrections when the reference genome and short-reads were aligned. A circulator (version 1.5.5) [36] was used on the genomes to change the start position to the *dnaA* gene (with –min_id 70 option) (available at GitHub https://github.com/CiriacC/Hybrid_bacterial_genome_assembly (accessed on 4 September 2022)). Assemblies’ statistics of new complete genomes were calculated using Quast (version 5.0.2) [37].

### 2.7. Genome Comparison

IS*6110*, IS*1081*, and IS*1561* insertion sequences were screened on genomes using Bionumerics (version 7.6.2) software created by Applied Maths NV and available from http://www.applied-maths.com (accessed on 10 June 2022). Genomes were aligned with progressiveMauve (version 2.4.0) [38] to determine the genomic structure. The list of indels was obtained using progressiveMauve for each complete genome in comparison to genome reference (Mb3601). In this study, we selected indels of at least 10 bp. Genes involved in these genetic events were inferred in comparison to the reference genome annotation (Mb3601). A comparison of these genetic events and the already known RDs was performed according to the Bespiatykh study [39].

### 2.8. Pangenomic Analysis

Genomic annotation was carried out with the Prokka (version 1.14.6) [40] tool using prodigal [41] and Mb3601 genbank file to predict ORF (open reading frame). 

The pangenomic study was performed with Panaroo (version 1.2.8) [42] with –merge_paralogue and –clean-mode strict options on 12 complete genomes (10 new complete genomes and 2 reference genomes). Visualization of the pangenome, core genome, and new genes accumulation curves were computed using the “gene_presence_absence. Rtab” matrix provided by Panaroo analysis and PanGP software [43] with Distance Guide sample Algorithm, 500 samples size, 100 samples repeat, and 100 amplification coefficients. Accessory genes were aligned with the reference genome (Mb3601) using blastn [44] to search for events that could explain their affiliation to the accessory genome.

### 2.9. Whole Genome SNP Identification and Selection

Genome reads were aligned to Mb3601 using BWA mem and samtools [45,46]. Vcf files were produced using bcftools mpileup and the SNP calling was made with bcftools call (with –vm option) [47]. The product was filtered using vcffilter and −f QUAL > 150 −f DP > 20 −f MQ > 49 options. All SNPs previously detected have been used to list all the variant positions of the panel. This list was used to make a second calling (using bcftools mpileup then bcftools call with -m option) only on these positions in order to have the same information for all strains and facilitate their fusion with bcftools merge. SNP annotation and effect prediction was performed using SnpEff and Mb3601 reference genome. The final steps of variant calling were performed with vcflib vcfsnps, vcflib vcffixup, vcflib vcfnumalt, and vcffilter (−f ‘NUMALT = 1’ option) [48]. SNPs supported by less than five reads forward and five reads reverse were filtered. Indel and SNPs with ambiguous nucleotide present on at least one strain have also been filtered. SNPs present in PE/PPE family protein and *pks12* were also filtered because of the low confidence and the higher error rate of these regions [48,49,50].

### 2.10. Phylogeny Based on SNP

Evolutionary trees were inferred on Mega [51] using the maximum likelihood method (Hasegawa–Kishino–Yano model) based on concatenated and validated SNPs (7023 SNPs for 98 genomes). The trees were drawn to scale, with branch lengths measured in the number of substitutions per site. Trees were midpoint rooted. A phylogenetic tree was visualized using the Interactive Tree of Life [52].

## 3. Results

### 3.1. Complete Genomes Features

For each of the 10 genomes, we obtained a complete assembly with 1 circular contig. Genomic characteristics are consistent with previous reference genomes and show great stability in genomic characteristics [18,24]. The genetic structure of the complete genomes has high stability (Appendix A). However, some differences are present especially in length and coding sequences (CDS) number (Table 1). Mb1855 has an addition of 23,948 bp and 48 CDS in comparison to Mb2377. 

All genomes have three rRNA and one tmRNA. Almost all genomes have 52 tRNA, one of them presenting a mutation in position 77 (C→T) in Mb3114, which has only 51. 

Insertion sequence (IS) analyses showed that all genomes have in the same position one copy of IS*1561* and six copies of IS*1081*, of which one is truncated (Table 1). According to our previous study [53], the number of IS*6110* is variable depending on the genotype. With 12 copies, Mb1855, which belongs to the Eu3 clonal complex, presents the highest IS*6110* (1355 bp) copy number, which is one of the main reasons for its large genome size. In Mb0820, in contrast not only to the other two Cluster A genomes Mb0531 and Mb0486 but also to the rest of the genomes belonging to other clusters, the otherwise ancestral recurrent copy of IS*6110* in the DR locus is absent. In Mb2377genome, representing Cluster G, there is a large deletion in the DR region which encompasses a portion of IS*6110* including orfA. Almost all IS*6110* except for the truncated copy in Mb2377, have a duplication of 2–4 bp in their insertion sites generated during IS transposition [54,55]. 

Together with the presence of IS*6110* variable copy numbers, genome size differences can also be explained by the presence of deletions or insertions (indels).

### 3.2. Pangenome Analysis and Gene Content Variation

Pangenome analysis on 12 complete genomes (10 described here, plus AF2122/97 and Mb3601) showed 3996 ortholog clusters and confirmed the high clonality of this species as regards the high core genome (98%, 3900 core genes) (Figure 1). The analysis showed 78 shell genes and 18 cloud genes (Appendix A).

Cloud genes are found in seven genomes that belong to three different clusters (A, C and I) (Figure 2). Cloud genes of Mb0486 and Mb0531 correspond to PE PPE genes and one hypothetical protein for Mb0486 (Appendix A). The cloud genes of Mb1101, Mb3114, and Mb3602 are annotated as hypothetical proteins. The cloud genes of Mb1855 and Mb3601 are due to IS*6110* insertion in the CDS except for *folp* found in Mb3601 (Appendix A). 

The low number of accessory (shell and cloud) genes is consistent with the alpha diversity of 1.11 which highlights a closed pangenome (Figure 3). In addition, a more detailed examination of accessory genes shows that some of them are present but pseudogenized (Appendix A). Indeed, 14 accessory genes have been listed due to an IS*6110* insertion which interrupts CDS. For example, the rpfD_2 orthologous gene present in Mb3601 is due to IS*6110* insertion, in *rpfD*. Fourteen other accessory genes were implied in the putative PhiRv1 phage protein (RD3) [39]. RD3 is present in the three complete genomes of Cluster A, Mb3602 (Cluster C), and Mb2377 (Cluster G). Two of the ortholog clusters present in RD3 are absent in Mb1101 but this genome has the other 12.

The other 16 accessory genes concern PE and PPE genes. These genes are known to be highly polymorphic and are often excluded from analyses [56]. They were excluded from our wgSNP study but not from the indel analysis. The region with the most indels found in our study, located at position 3,890,000 bp in Mb3601 genome, encompasses PE PGRS genes (pe_pgrs 59, pe_pgrs 54, pe_pgrs 56 and pe_pgrs53). Other regions presenting numerous indels are CRISPR-Cas (position 3,090,000 bp in Mb3601) or a region including PPE genes (position 2,165,000 bp in Mb3601). This result shows that the indel distribution is not random (Appendix A).

The comparison of complete genomes against Mb3601 as reference genome shows 72 indels for Mb2487, 56 for Mb1101, 54 for Mb1855, 34 for Mb3114, 74 for Mb0820, 88 for Mb0531, 83 for Mb0486, 58 for Mb2377, 77 for Mb2269, and 69 for Mb3602 (Appendix A).

Some large indels (more than 2 kb) were identified in the complete genomes (Table 2 and Appendix A).

### 3.3. Contribution of the Complete Genome to M. bovis Lineages Definition

Obtaining complete genomes was an opportunity to revisit the population structure of French *M. bovis* strains by looking at the topology of the SNP-based phylogenetic tree and identifying genetic traits that could complete the new nomenclature covering the main *M. bovis* phylogenetic groups [11]. 7023 SNPs were found among the 98 genomes (12 complete and 86 draft genomes) (Appendix A). The majority (87.7%) of SNPs were present in CDS and 12.3% are intergenic. The analysis showed that 31.4% are synonymous variants and 56.3% are non-synonymous variants. The phylogenetic distribution of SNPs, shown in Figure 4, discriminated *M. bovis* genomes into 10 clusters well resolved by at least 200 SNP. This population structure is congruent with previous studies [7,53]. Indeed, the heatmap clearly highlights lineages based on the absolute SNP distance between strains, which supports a very clear separation between the lineage La1.2 (clusters G+H+I) and lineage La1.8.2 (cluster A+B+C). However, lineages 1.7 or 1.8 are not clearly identified in this figure.

All groups described in this study have specific SNPs (Figure 5). Some groups have few specific SNP such as Cluster C, and Eu3. Other groups have more than 60 specific SNPs such as Eu2, Cluster A, or Cluster G.

Indels were also examined on the 98 genomes and the specificity of an indel for an *M. bovis* group was determined when it was identified in all genomes of this group (Figure 5). 

#### 3.3.1. Cluster A/F4 Family

Three complete genomes were obtained for Cluster A: Mb0486, Mb0820, and Mb0531. This cluster is described by 66 specific SNPs and 8 specific indels (Appendix A). In comparison to Mb3601, the deletions involve *metk* (MBS3601_RS07300) and a *leuA* (MBS3601_RS19090) partial deletion. However, *leuA* was also partially deleted in the different and larger indel of Mb2487 (cluster F). Two recurrent IS*6110* insertions sites were found in the three complete genomes. Other genomic characteristics of this cluster are the absence of spacer 33 in the DR region, RD3, and the truncated repetition of QUB26. 

#### 3.3.2. Cluster C/SB0134 Family

This cluster is composed of two subgroups and is described by few SNP and only one deletion of 514 bp (Figure 4 and Figure 5). This group does not present spacers 4 and 5 in their spoligotypes. Mb3602 and Mb2269 are present in each of these subgroups. 

#### 3.3.3. Cluster F/Eu2

Mb2487 is representative of this clonal complex, which is defined by 77 SNPs, including that in *guaA* described originally [3] and a lack of spacer 21 in their spoligotypes. 

#### 3.3.4. Lineages 1.7 and 1.8

A deletion of 2409 bp (Indel-Mb0486-49, Indel-Mb0531-56, Indel-Mb0820-44, Indel-Mb3602-44, Indel-Mb2269-47, and Indel-Mb2487-50) which corresponds to RDBovis [39] is common to genomes of cluster A, C, and F and allow to define the lineage La1.7 + La1.8. This lineage is also defined by 108 SNPs and an insertion of more than 2000 bp (Indel-Mb2487-64, Indel-Mb2269-63, Indel-Mb3602-58, Indel-Mb0820-65, Indel-Mb0531-76, and Indel-Mb0486-49). However, the insertion present in Mb2487 is the largest compared to those in the other complete genomes. This region contains PPE genes. These two indels are also present in AF2122/97, as shown in a previous study comparing this genome to Mb3601.

#### 3.3.5. Cluster G/F9 Family

Mb2377 is representative of cluster G, which is defined by 83 SNPs. As mentioned before, this cluster is characterized by the truncated IS*6110* in the DR region and the lack of spacer 1 to 17. These specific genetic characteristics are due to a large indel (Indel-Mb2377-27). 

#### 3.3.6. Cluster I/Eu3

In addition to Mb3601, three complete genomes were obtained for this cluster, which is the most represented among the strains studied in France [12,14]. The Eu3 clonal complex is only defined by two SNPs. Indeed, Mb1101 is close to BCG vaccine strains and is separated from other Eu3 strains (Figure 6). 

We propose to define Cluster I1 which corresponds to Cluster I strains by removing vaccine strains and Mb1101. Mb3114, Mb3601, and Mb1855 are present in this cluster, which is defined by 53 SNPs and 3 indels. One of these indels, a deletion of 622 bp, includes *VapB46* (Appendix A).

## 4. Discussion

Using Illumina and MinION sequencing technologies, 10 complete genomes of *M. bovis* were obtained and represented—with AF2122/97 and Mb3601—the main *M. bovis* clusters described previously [7]. Our analysis showed highly similar genomic features and a conserved synteny within these new 10 complete genomes. Pangenome established with the ten complete genomes and the two other available complete genomes [18,24] confirmed a closed pangenome with a core gene representing 98% of total genes in agreement with previous studies [4,15,57]. However, a recent study using the “Get-homologues pipeline” and draft genomes showed an open pangenome and a larger accessory genome in comparison to our study [58,59]. This difference can be explained by the short-read sequences data, which lead to the increase of the accessory genome [15]. To overcome this problem, Panaroo can be used to clean up annotation errors due to fragmented assemblies or misassembly [42]. Indeed, Panaroo produces superior ortholog clusters, which induce a reduction in the accessory genome estimate size and an increase in the core genome. The pangenomic analysis of 12 complete genomes highlighted an alpha diversity of 1.11 consistent with a closed pangenome [60]. In addition, the presence or absence of certain ortholog clusters in genomes is due to gene pseudogenization. Our analysis showed that the size of the core genome decreases more rapidly than the increase in the pangenome size corroborating that evolution of the MTBC complex members genomes, as recently demonstrated for *M. bovis* [61], occurs by gene loss or pseudogenized instead of gene gain. This event could explain the pathogen’s host specialization as shown in *M. tuberculosis* [62,63]. 

Among genomic features, we observed variations in genomes size between the 10 complete genomes. This observation was explained by a variable number of copies of IS*6110* according to the genomes and the indel content. Indeed, the 12 copies of IS*6110* in Mb1855 represent an addition of 14,905 bp in comparison to genomes with only one copy of IS*6110*. The complete genomes allowed us to confirm the presence of multiple copies of IS*6110* in certain *M. bovis* genotypes according to our previous study [24,53]. The transposition of this genetic element can play an important role in bacterial evolution by interrupting or leading to the overexpression of genes [53,64,65,66]. Indeed, some genetic changes such as gene deletion or gene pseudogenization that could affect the core genome, can be attributed to IS*6110*. Multiple examples are present in literature and show the deletion of some genes like *cas* genes in the CRISPR-Cas locus [65]. In our study, one of these examples is present in Cluster G strains with the absence of *Cas* genes and the first 17 spoligotype spacers. However, except for this example, all IS*6110* have a duplication of 2–4 bp in their insertion sites which in the nine other complete genomes shows the absence of IS recombination events between two IS*6110*.

Indels can also explain length differences among genomes. Some large deletions are identified in this study as Indel-Mb2377-27 of 5539 bp in Mb2377, Indel-Mb2487-64 of 5166 bp in Mb2487, RDBovis of 2409 bp present in genomes of La1.7 and La1.8 or RD3 of 9253 bp present in Cluster A and several other genomes [39]. This last indel corresponds to prophage phiRv1 which seems to have a role in host hypoxia [61,67,68]. However, Mb1101 has a specific deletion pattern in this region that involves two ortholog clusters instead of 14 in RD3. In addition, our results showed that indel positions are not random. Many indels are present in the CRISPR-Cas region [65] but the most polymorphic region is that containing PE and PPE genes. This high frequency of deletions and insertions in these regions is in agreement with the previous *M. bovis* complete genome publication [24]. Further studies on these indels are needed to better understand their role in bacterial evolution.

In this study, the selected *M. bovis* strains to obtain complete genomes, represent the main genotypes responsible for TB outbreaks in France and are also representative of *M. bovis* genotypes found in other countries. Indeed, Mb2487 belongs to the lineage 1.7.1, formerly described as Eu2 clonal complex [3,8,11]. Four complete genomes belong to lineage La 1.2, 3 of which belong to the Eu3 clonal group (in addition to Mb3601). Mb1855 is representative of highly prevalent strains in France with several copies of IS*6110*. Mb3114 is representative of a common genotype in Italy with only one copy of IS*6110* [69]. Five genomes belong to lineage 1.8.2. This lineage had previously been separated in the Hauer study into Cluster A, B, and C [7]. Harmonization of the nomenclature used to describe *M. bovis* lineages may facilitate comparisons of WGS studies. Specific indels and SNPs were described for complete genomes or *M. bovis* lineages. Some of these genetic events such as *guaA* and other 68 SNPs specific to the Eu2 strains [10], already described in the literature, were confirmed in this study. Nevertheless, the number of specific SNPs found for the previously described clusters was larger than what was found in a recent study [11]. These differences can be explained by the smaller number of strains used in our study. This result shows the importance of using a panel of strains as exhaustive as possible to describe specific events of the *M. bovis* lineage. Some indels were found to be specific to *M. bovis* lineages, others appear to be specific to certain genomes. They will need to be investigated in larger panels of strains to determine if they are the signature of groups or subgroups of *M. bovis*.

TB cattle outbreaks in France are present in specific regions where *M. bovis* circulates in wild and domestic communities of hosts [13,25] where the transmission links between infected animals remain difficult to establish as *M. bovis* strains share spoligotype and a multilocus variable number of tandem repeats analysis (MLVA) identical profiles [70,71,72]. WGS-SNP can be used to refine these studies but requires adapted reference genomes to the field strains. Mb3601 and other representative complete genomes could be used to improve epidemiological studies for the surveillance of TB and contact tracing between infected animals [16]. The new complete genomes described in this study are closer to field strains than AF2122/97, the genome used as a reference until now, which will allow better epidemiological surveillance of the disease based on WGS data.

## 5. Conclusions

Ten new *M. bovis* complete genomes were obtained in this study. These new complete genomes cover the *M. bovis* French diversity but are also representative of *M. bovis* lineages present in other countries. These genomes allow us to better describe *M. bovis* lineages. A comparison of these complete genomes confirmed that the global genome organization of *M. bovis* is very stable and shows a closed pangenome. The search for indels and SNPs made it possible to specify certain genomic traits and the absence of certain genes characterizing each cluster described in this article.

These complete genomes, adapted to *M. bovis* clusters, will be useful to better understand TB transmission dynamics in multi-host systems and therefore to implement more effective control measures.

## Figures and Tables

**Figure 1 microorganisms-11-00177-f001:**
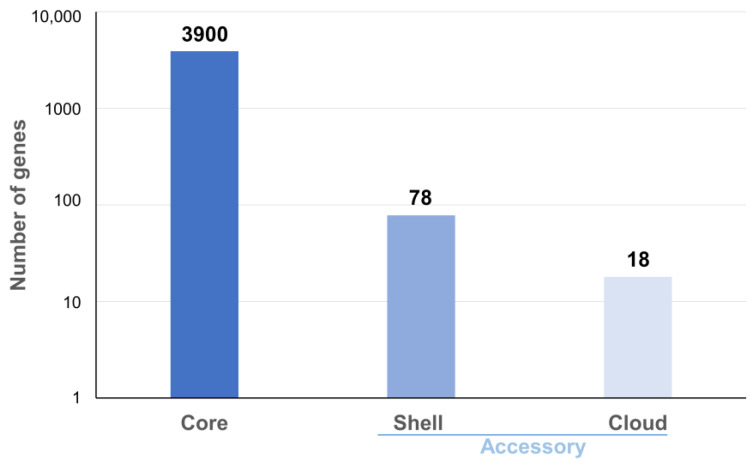
Pan-genomic histogram of 12 complete genomes of *M. bovis*. The figure shows the core and accessory genes proportion in the genome’s panel.

**Figure 2 microorganisms-11-00177-f002:**
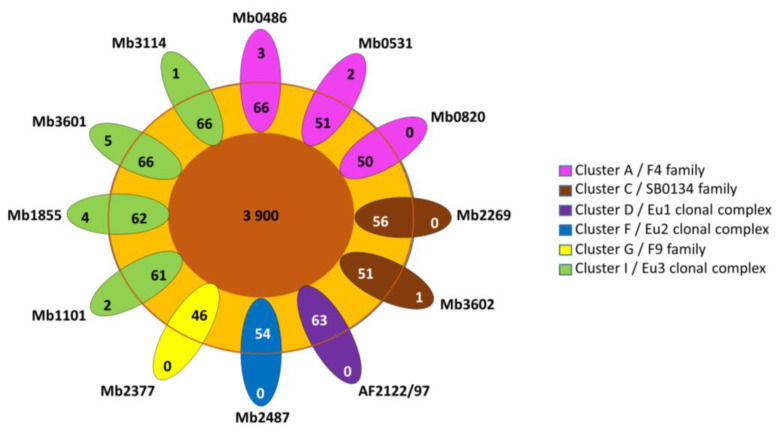
The gene distribution of pangenome. Flower plot showing in the center, genes present in all strains (core-genes), genes present in some strains (shell genes) in the annulus, and strain-specific genes of the 12 *M. bovis* complete genomes in the petals (cloud genes). Genomes are grouped in 6 previously described clusters [7].

**Figure 3 microorganisms-11-00177-f003:**
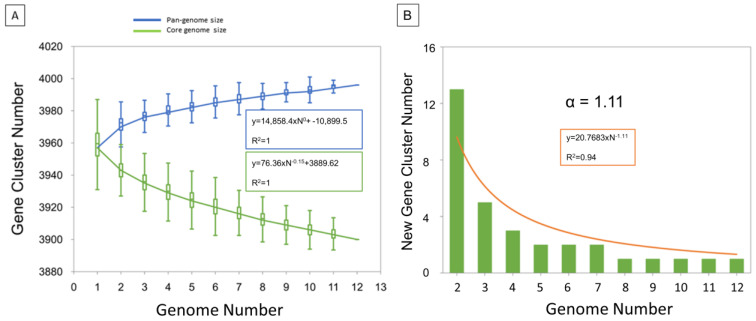
Pan-genome profile calculated with PanGP tool. (**A**) The figure shows two gene cluster accumulation curves for pangenome (blue) and core genome (green). (**B**) Evolution of new gene cluster numbers over genome number. The trend line (in orange) defines the curve equation and the alpha diversity.

**Figure 4 microorganisms-11-00177-f004:**
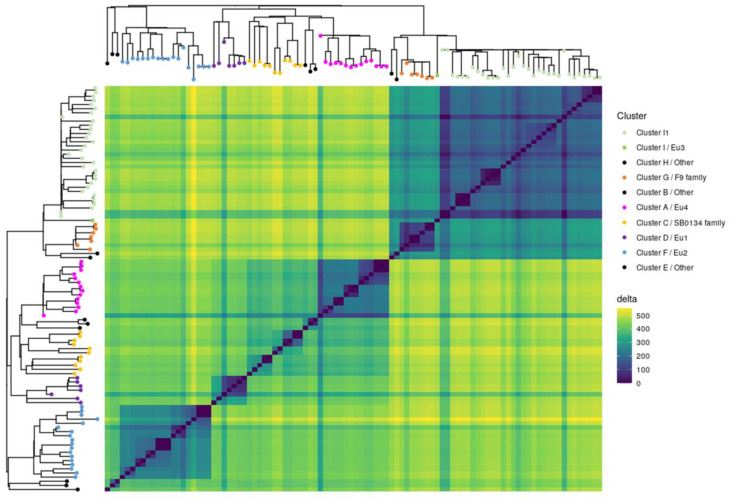
*M. bovis* isolates separated into clusters. The heatmap illustrates pairwise SNP distance between genomes belonging to each cluster. Both axes have a maximum-likelihood SNP-based tree inferred on 98 genomes with leaf colored according to cluster defined in this study. Trees were midpoint rooted. The SNP difference key is shown on the right.

**Figure 5 microorganisms-11-00177-f005:**
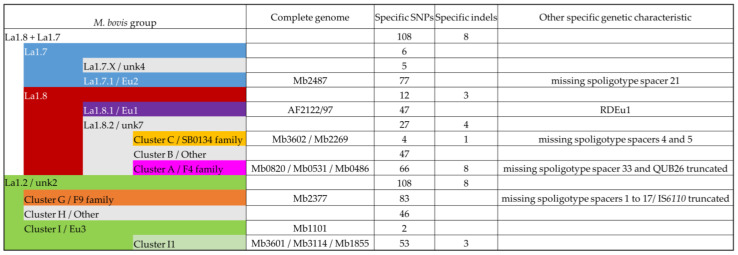
List of specific genetic events of the different *M. bovis* groups (Appendix A). The colors of the *M. bovis* groups are in accordance with the previously described clusters and lineage Hauer et al. 2019 [7].

**Figure 6 microorganisms-11-00177-f006:**
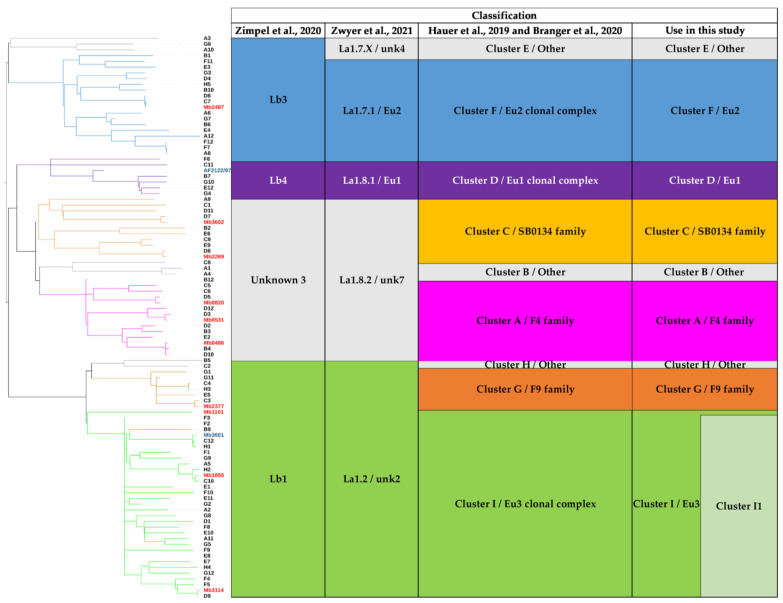
Phylogenetic tree of 98 *M. bovis* genomes. The two previous reference genomes (Mb3601 and AF2122/97) are marked in blue. The 10 new complete genomes are indicated in red. The phylogenetic tree is based on 7023 whole genome SNPs. The strains are grouped according to the previous classification Hauer et al. 2019, Zwyer et al. 2021 and Guimares et al. 2020 [7,11,16]. The colors of the *M. bovis* groups are in accordance with the previously described clusters and lineages Hauer et al. 2019, Zimpel et al. 2020 and Zwyer et al. 2021 [7,10,11].

**Table 1 microorganisms-11-00177-t001:** Information on the 10 *M. bovis* strains selected and sequenced in this study.

Name	Mb2487	Mb3602	Mb2269	Mb0820	Mb0531	Mb0486	Mb2377	Mb1101	Mb1855	Mb3114
Accesion Number	CP096839	CP096843	CP096840	CP096841	CP096847	CP096848	CP096846	CP096845	CP096844	CP096842
Host species	Cattle	Deer	Cattle	Cattle	Cattle	Cattle	Cattle	Cattle	Cattle	Cattle
Spoligotype ID	SB0999	SB0134	SB0134	SB0840	SB0826	SB0821	SB0853	SB0120	SB0120	SB0120
MLVA profile *	6 4 5 2 8 2 4 7	7 4 5 3 10 4 5 10	6 5 5 3 6 4 5 6	7 5 5 3 8 2 5 s 4	6 7 3 3 10 2 5 s 8	6 5 5 3 11 2 5 s 4	3 6 5 2 9 3 4 6	5 2 3 3 10 3 3 10	5 3 5 3 9 4 5 6	5 5 5 3 11 3 5 4
Cluster	F	C	C	A	A	A	G	I	I	I
Alias	Eu2 CC	SB0134 family	SB0134 family	F4 family	F4 family	F4 family	F9 family	Eu3 CC	Eu3 CC	Eu3 CC
Lenght (bp)	4,344,516	4,343,218	4,351,057	4,344,564	4,342,977	4,340,629	4,338,946	4,343,846	4,362,894	4,353,147
GC (%)	65.62	65.65	65.64	65.64	65.64	65.65	65.65	65.64	65.64	64.65
CDS	4012	3999	4014	4006	4005	3991	3986	4014	4034	4015
rRNA	3	3	3	3	3	3	3	3	3	3
tRNA	52	52	52	52	52	52	52	52	52	51
tmRNA	1	1	1	1	1	1	1	1	1	1
IS*6110* Nb	3	1	3	2	4	3	1 truncated	1	12	1
IS*1561* Nb	1	1	1	1	1	1	1	1	1	1
IS*1081* Nb	5 + 1 truncated	5 + 1 truncated	5 + 1 truncated	5 + 1 truncated	5 + 1 truncated	5 + 1 truncated	5 + 1 truncated	5 + 1 truncated	5 + 1 truncated	5 + 1 truncated

* MLVA *loci*: ETR A, ETR B, ETR C, ETR D, QUB 11a, QUB 11b, QUB 26, QUB 3232.

**Table 2 microorganisms-11-00177-t002:** Details of large indels affecting the genomes.

Nomenclature	Length (in bp)	Number of Locus Tags Associated	Genome
Indel-Mb0531-33	2384	5	Mb0531
Indel-Mb0486-6	3148	4	Mb0486
Indel-Mb0486-11	3634	4	Mb0486
Indel-Mb3602-33	2150	2	Mb3602
Indel-Mb2269-1	2122	4	Mb2269
Indel-Mb2269-24	2368	6	Mb2269
Indel-Mb2487-5	2691	3	Mb2487
Indel-Mb2487-36	2387	6	Mb2487
Indel-Mb2487-50/RDBovis	2409	3	Mb2487
Indel-Mb2377-27	5539	6	Mb2377
Indel-Mb1101-1	2966	2	Mb1101
Indel-Mb1101-8	4384	2	Mb1101
Indel-Mb1101-21	1160	1	Mb1101
Indel-Mb1855-26	1730	1	Mb1855
Indel-Mb1855-29	3058	3	Mb1855

## Data Availability

The raw data are deposited in a public domain server at the NCBI SRA database, under BioProject accession number PRJNA832544.

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
