# Peer review of "Features of Mycobacterium bovis Complete Genomes Belonging to 5 Different Lineages"

_microorganisms, 2023, doi:10.3390/microorganisms11010177_

Round 1

Reviewer 1 Report

This manuscript reports the complete genome sequencing of ten M. bovis strains representing each genetic lineage identified in France and/or in other countries and provides valuable genomic traits that allow us to better describe M. bovis lineages. The availability of new complete representative genomes of M. bovis will be useful to epidemiological studies and better understand the transmission of this clonal-evolving pathogen.

Few minor comments are included below in order to improve the manuscript:

General comments

·  The authors reported features of M. bovis complete genomes belonging to 5 different lineages (clusters A, C, F, G, and I), being the most prevalent in the last 20 years in France the A, C, and I clusters. Is there any additional explanation for not including other strains belonging to the rest of clusters detected in France (B, D, E, and H)?

·         The term bovine tuberculosis should be restricted to the infection by M. tuberculosis complex in bovine animal species and in this manuscript, it is used for the infection in cattle and other animal species. Please, revise the document and consider the proposal as a modification.

M&M.

·         2.1 Mycobacterium bovis should be in italics.

Results.

·         Line 208. Please, include in brackets the 3 different clusters with cloud genes (A,C, I)

·         Lines 208-2012. Seven genomes include cloud genes. The authors mentioned if they belong to specific proteins/genes in six of them (Mb0486, Mb0531, Mb1101, Mb3114, Mb1855, Mb3601) but did not inform about the Mb3602. Any reason about this?

·         Lines 245-249. The number of indels per genome is detailed. How many indels are in common between the genomes or are exclusive for any genome? Is it relevant?

·         3.3. M. bovis should be in italics

·         Lines 269-271. 87.8% of SNPs are present in CDS…. But the analysis showed 31.4 and 56.3% synonymous and non- synonymous variants, respectively. 31.4 plus 56.3 is 87.7%. Could you adjust the numbers to 87.8% of the SNPs at the CDS regions?

Tables and Figures

·         Table 1. Specify as a footnote the order of the VNTR loci.

Minor comments

·         Check italics when applied in all the bibliography.

Author Response

Point-by-point response to reviewer comments for Manuscript ID: microorganisms-2089475:

Reviewers' comments:
Reviewer #1:
Reviewer comment 1: The authors reported features of M. bovis complete genomes belonging to 5 different lineages (clusters A, C, F, G, and I), being the most prevalent in the last 20 years in France the A, C, and I clusters. Is there any additional explanation for not including other strains belonging to the rest of clusters detected in France (B, D, E, and H)?

Authors' response: Thank you for the comment. Indeed, the reason on why not having analysed additional genomes of strains belonging to clusters B, E, and H is the lack of outbreaks due to these strains in the last two decades. Cluster D correspond to European clonal complex for which there is already a reference genome (AF2122/97).

Reviewer comment 2: The term bovine tuberculosis should be restricted to the infection by M. tuberculosis complex in bovine animal species and in this manuscript, it is used for the infection in cattle and other animal species. Please, revise the document and consider the proposal as a modification.

Authors' response: We have changed bovine tuberculosis (bTB) by mammalian tuberculosis (TB) as per in the new version ad hoc chapter of the WOAH Mammalian terrestrial manual.  

Reviewer comment 3: M&M.· 2.1 Mycobacterium bovis should be in italics.

Authors' response: Indeed, in the format of the journal the subtitles of section are in italic and in this case the writing font is reversed hence the fact that “Mycobacterium bovis” is not in italic in this subtitle.

Reviewer comment 4: Results.
-Line 208. Please, include in brackets the 3 different clusters with cloud genes (A,C, I)

Authors' response: According to your recommendations, we have added this information in the revised version.

Lines 208-2012. Seven genomes include cloud genes. The authors mentioned if they belong to specific proteins/genes in six of them (Mb0486, Mb0531, Mb1101, Mb3114, Mb1855, Mb3601) but did not inform about the Mb3602. Any reason about this?

Authors' response: We agree with the reviewer, this missing information has been added in this revised version. The Mb3602 cloud gene is annotated as hypothetical protein. We have modified the sentence, line 210, as follow: The cloud genes of Mb1101, Mb3114 and Mb3602 are annotated as hypothetical proteins.

Lines 245-249. The number of indels per genome is detailed. How many indels are in common between the genomes or are exclusive for any genome? Is it relevant?

Authors' response: Most indels are common within each cluster. Table S3 shows the details of this information. In this revised version we have added the legend in the figure of table S3 "indel distribution". Regarding the relevance of indels, ideally data should be available for all strains to see if the indel-based phylogenic tree could basically reflect the phylogenetic relationships of clusters.

3.3. M. bovis should be in italics

Authors' response: As mentioned above, this is due to the format of the journal where the subheadings of the sections are in italics.

Lines 269-271. 87.8% of SNPs are present in CDS…. But the analysis showed 31.4 and 56.3% synonymous and non- synonymous variants, respectively. 31.4 plus 56.3 is 87.7%. Could you adjust the numbers to 87.8% of the SNPs at the CDS regions?

Authors' response: Thank you for pointing out this error. We have corrected it to 87.7 in this revised version.

Reviewer comment 5: Tables and Figures

Table 1. Specify as a footnote the order of the VNTR loci.

Authors' response: This has been done in the revised manuscript.

Reviewer Minor comments 1:
Check italics when applied in all the bibliography.

Authors' response: In this revised version we have checked and modified where necessary the italics in the bibliography

Reviewer 2 Report

Reviewer comments to authors:

The manuscript entitled “Features of Mycobacterium boivs complete genome belonging to 5 different lineages”, acquired ten novel M. bovis genomes this study. These new full genomes represent not only the French diversity of M. bovis, but also the diversity of M. bovis lineages in other countries. These genomes enable us to better describe the lineages of M. bovis. Comparison of these full genomes demonstrated that the global genomic organisation of Mycoplasma bovis is extremely stable and demonstrates a closed pangenome. The evaluation for indel mutations and single nucleotide polymorphisms enabled the identification of specific genomic characteristics and the lack of certain genes for each cluster described in this study. The study is intriguing and relevant to the journal's topic. The manuscript is well-written, and the data analysis and results interpretation are organised logically. I would thus recommend that the manuscript be considered after minor revisions.

General Questions

- Incorporate accession number into hyperlink?

- Do clonal strains have more indels than non-clonal strains?

- Will the indels give insight into the pan and core genome and potential gene repertoires that may determine tissue or animal species tropism?

 Minor comments

L43: Short read or short-read, confirm throughout the text

L45: species in eight or species into eight

L49: cluster C provoke or cluster C provoked

L63: specific of or specific to

L66: leaded or led

L76: Ten strains mentioned the details in Table number?

Table S2. What is Feuil 1?

Author Response

Point-by-point response to reviewer comments for Manuscript ID: microorganisms-2089475:

Reviewer #2:
Reviewer General Questions

- Incorporate accession number into hyperlink?

Author response: According to your recommendation, we have added the hyperlink line 455 but The WGS data has been deposited on NCBI with a release date and will be accessible after this article is published, so the hyperlink is not active at this time.

- Do clonal strains have more indels than non-clonal strains?

Author response: It is difficult to answer this interesting question. The abundance and evolution of indels in bacterial genomes are still very poorly documented. Most population genomic studies in bacterial genomes relied on SNPs, because accurate inference of indels from next-generation sequencing data is still challenging. It is possible that like for SNPs, the number of indels in non-clonal genomes is greater.

- Will the indels give insight into the pan and core genome and potential gene repertoires that may determine tissue or animal species tropism?

Author response: The reasons that may determine a host or tissue tropism and virulence are difficult to explain, but it is probably a combination of factors related to the host and the bacterium.

M. bovis strains evolves clonally, and their genomes are not subject to horizontal gene transfer or significant recombination events. M. bovis, like members of the M. tuberculosis complex, shape their genetic content primarily through gene loss and duplication events. Despite low genetic variability, these microorganisms evolve through large deletions (some called regions of difference, "RDs"), SNPs, short indels, duplication of a limited number of paralogous gene families, and transposition of insertion sequence elements.
Indels can substantially affect gene function by disrupting the reading frame of a gene but little is understood about the fitness effects of indels, their distribution in the genome, and their role in adaptive evolution. Underlying host tropism and virulence are elusive but likely a combination of host and bacterial factors.
How these indels lead to changes in the adaptive characteristics of M. bovis is not known.

Reviewer Minor comments
L43: Short read or short-read, confirm throughout the text

Author response: Thank you for pointing this out, we have harmonized the writing “short-reads” in this revised version.

L45: species in eight or species into eight

Author response: correction done

L49: cluster C provoke or cluster C provoked

Author response: correction done

L63: specific of or specific to

Author response: correction done

L66: leaded or led.

Author response: correction done

L76: Ten strains mentioned the details in Table number?

Author response: According to this remark, we have modified this sentence and added "in Table 1"

Table S2. What is Feuil 1?

Author response: sorry for this oversight, we have modified by "Sheet 1" in this revised version